# Research on Subdivision System of Sin-Cos Encoder Based on Zero Phase Bandpass Filter

**DOI:** 10.3390/s19143041

**Published:** 2019-07-10

**Authors:** Haoning Zhao, Jiazhong Xu, Haibin Zhang, Zhen Liu, Shi Dong

**Affiliations:** 1School of Automation, Harbin University of Science and Technology, Harbin 150080, China; 2HRG Hefei International Institute for Research and Innovation, Hefei 230000, China; 3Hit Robot Group, Harbin 150001, China; 4State Key Laboratory of Mechanical System and Vibration, Shanghai Jiao Tong University, Shanghai 200240, China

**Keywords:** Sin-Cos encoder, software subdivision, zero phase bandpass filter, high-speed sampling, stream processing

## Abstract

A novel high-precision subdivision system for high-speed encoders is designed in this work. The system is designed with an arc second of Sin-Cos Encoder (SCE) based on zero phase bandpass filter. The system collects the analog output signals of an encoder with a high-speed data acquisition system (DAS); the noise of a digital signal can be effectively eliminated by zero phase bandpass filter with appropriate prior parameters. Finally, the actual rotation angle of the encoder is calculated by the software subdivision technique in the system. The software subdivision technique includes two methods, which are the Analog Pulse Counter (APC) and the Arc Tangent Subdivision (ATS). The APC method calculates the encoder angle by counting the analog pulses acquired by the arc tangent signal. The ATS method calculates the encoder angle by computing the arc tangent results of each point. The accuracy and stability of the system are first verified with a simulated signal; second, the real signals of an SCE are acquired by a high speed DAS on a test bench of a precision reducer, which is employed in industrial robots. The results of the proposed system are compared. The experimental results show that the system can significantly improve the accuracy of the encoder angle calculation, with controllable costs.

## 1. Introduction

In precision measurement and control systems with high dynamic and precision performance requirements, the position and rotational speed of the rotator must be accurately measured and perceived. The high-precision Sin-Cos Encoder (SCE) is a kind of sensor that converts the geometric displacement of the rotating shaft into a digital quantity [1]. Based on the digital quantity, the rotator position and speed detection are realized by alternative technologies. The high-precision SCE is widely used in precision reducers with high-precision detection, robot technology, precision machine tools, and other fields [2]. Especially with the increasing precision requirements of measurement and control systems, and under the circumstances of physical accuracy limitations of the encoder, the high-speed high-precision signal acquisition and data processing technologies are of great importance. Currently, the precision reducer comprehensive performance testing equipment, the reducer input, and the output of the angle measurement accuracy requirements have reached 1; the output speed has reached 2000 rpm. Addressing these demanding specifications by increasing the number of encoder gratings, the costs and the technology constraints imposed by the existing encoder and subdivision technologies, is not feasible. High-precision test data is an important guarantee of the reducer testing system. The signal acquisition system of the reducer is an important part of the reducer testing. It is closely related to the accuracy of the test data of the reducer [3]. In order to improve the resolution and precision of the encoder and reduce the hardware costs, this paper proposes a high-precision encoder actual rotation angle method. The method adopts a high-speed sampling of an encoder signal (over 40 MHz), filtering processing, as well as software segmentation. Finally, the actual rotation angle of the high precision encoder is obtained.

The encoder subdivision methods of the Moire fringe can be in organized into three categories: optical, mechanical, and electronic [4]. Among them, the electronic subdivision methods have the advantages of high precision, ease of automation, etc.; it has been widely used. The commonly used electronic subdivision category is divided into hardware methods (i.e., circuit realizations) and software methods (i.e., programming realizations) [5]. Chen et al. [6] propose the phase coding subdivision technique of the incremental encoder, which improves the resolution by encoding the phase of two sinusoidal signals with a 90° phase difference from the encoder output. He et al. [7] propose a new high-magnification encoder subdivision method to improve the accuracy with a nonlinear serrated wave compensation, which is suitable for DSP controller implementations. Wu et al. [8] propose the encoder subdivision technology based on a closed-loop tracking method, which is suitable for ARM processor implementations. Feng et al. [9] propose the error correction method of the fine code Moire fringe subdivision of small photoelectric encoders, which can effectively improve the accuracy of the encoder without increasing the volume of the existing small photoelectric encoder processing circuit. Wu [10] proposes the encoder subdivision technology based on the CORDIC algorithm, which is suitable for FPGA processor implementations. Song et al. [11] present a method of encoder subdivision based on the software frequency multiplication, which narrows the data range and improves the operation accuracy by the method of proportional scaling of sin-cos signals. Liu et al. [12] present a method of using MATLAB software to realize encoder subdivision, which is suitable for PMSM motor control models. Wei et al. [13] present a high-resolution signal processing method of the SCE based on an improved coordinate rotational digital calculation algorithm.

It can be observed that in the current cosine encoder high-speed signal subdivision methods, the hardware subdivision methods achieve subdivision through the comparator circuit, which has the disadvantages of complex circuits, high cost, and poor flexibility. The software subdivision methods generally apply DSP or FPGA processors in the existing literature. These convert the encoder’s sin-cos signal into an AD sampling and conversion. Affected by the speed of operation, it can only subdivide according to the amplitude or the phase of the table subdivisions. Furthermore, the software methods cannot meet the real-time requirements of the test system for high-speed signal low-magnification subdivision requirements, due to the weak pulse processing capacity. In addition, the mechanical jitter caused by the operation of the motor is also extremely challenging, and it impacts the kind of filtering technology to adopt to remove noise. The accuracy of the calculation for the actual rotational speed of the rotor also needs to be further improved. In the case of low real-time demands, the software subdivision system in this paper uses the industrial computer as the upper computer processor of the subdivision operation. These computers can directly perform the advanced operations and subdivision of the sin-cos signal according to the algorithm without checking the table. The accuracy is higher and can support an effective filtering algorithm, such as the zero phase bandpass filtering algorithm, to remove noise.

## 2. Sin-Cos Encoder and Subdivision Principle

### 2.1. Sin-Cos Encoder

The operational principle of the SCE is similar to a common photoelectric encoder [14]. The encoder rotates one circle to output thousands of cycles of a sin-cos signal and a zero point signal. The number of periodic signals is related to the number of engraving lines. The ordinary photoelectric encoder outputs multiple square wave signals; however, the SCE through subdivision can obtain a finer resolution than the pulse [15]. As a special incremental encoder, the SCE can achieve high precision without the density of engraving, and the detection resolution can be obtained more finely than the original signal period by the electronic subdivision of the sin-cos signal [16].

According to the Shannon sampling theory [17], the sampling frequency at the maximum rotational speed must exceed two times the frequency of the encoder output sin-cos signal. When we process the encoder output signal with a high sampling frequency and a large amount of data, we can collect processing information efficiently and quickly. Parallel streams processing technologies that the high-speed streams of large file data import disks are adopted to achieve this.

The sin-cos encoding signal refers to the encoder signal that loads the position information with two phase sine signals, which have difference of 90° [18]. A photoelectric scanning raster structure is usually used to produce such a signal. The LED light source and the focusing mirror provide a light source for stabilizing the grating diffraction stripes. The grid structure of the scanning mask and the measuring datum are the same or similar. When the slit is aligned, the timeline cannot pass through the alignment of the two lines. The light intensity sensed by the two relative moving photovoltaic cells exchanges; the light is a triangular waveform. However, because the diffraction efficiency of the grating is in the form of a sine wave, the photovoltaic cell converts this form of the optical signal into an electrical output [19].

### 2.2. Subdivision Principle

Ideally, the SCE outputs the two phases of an orthogonal voltage signal in a one-cycle rotation (phases A and B), as shown in Figure 1.

The voltage signals of phase A and phase B can be expressed as follows.
(1)$$U_{A}\left(t\right)=U\sin\left(\omega t+\varphi_{a}\right)$$
(2)$$U_{B}\left(t\right)=U\cos\left(\omega t+\varphi_{a}\right)$$
where U is the SCE output voltage amplitude, ω is the angular frequency, $\varphi_{a}$ is the initial phase, and t is time.

The principle of a software subdivision method can be expressed as follows; by selecting a point voltage $U_{p}$ in Equations (1) and (2) as the voltage reference point of output counting pulse, the count pulse is output when the input amplitude is $U\geq U_{p}$. When different reference voltages are selected, the encoder rotates over a certain angle, outputs a fixed pulse, and subdivides the sin-cos signal [20].

An alternative software subdivision method is to sample the two sin-cos signals, and calculate the ratio of the two signals in order to obtain the phase value in a direct way. Theoretically, this method can calculate any position of the motor rotor. Moreover, the accuracy can theoretically achieve the actual physical precision of the encoder.

## 3. System Composition and Algorithm Implementation

### 3.1. System Composition

The subdivision process of high-precision sin-cos encoding signal based on zero phase bandpass filter includes high-speed digitizer acquisition, zero phase bandpass filter, high-speed flow disk storage, and software segmentation. These elements, and the algorithm flow, are shown in Figure 2. The algorithm steps are as follows.
(1)High-speed data acquisition system (DAS) collects signals.(2)Zero phase bandpass filtering algorithm removes the noise.(3)High-speed stream processing.(4)Software subdivision, including Analog Pulse Counter (APC) and Arc Tangent Subdivision (ATS), and using a parallel stream processing mode.

### 3.2. Zero Phase Bandpass Filter

The zero crossings can be estimated and hence the speed. So we can simply apply zero phase bandpass filter to the signal and then extract the phase using the Hilbert transform without any noise. Usually, the signal will have phase shift after filtering in time domain. Zero phase filtering is a digital filtering technology to solve this problem. The phase response of zero phase filter system function is zero. That is to say, there is no phase deviation after filtering [21]. Zero phase filtering is an excellent solution to the signal processing process which needs to keep the signal phase after filtering [22]. Zero phase filtering is an excellent solution to the signal processing process which needs to keep the signal phase after filtering. Note that every even signal is symmetric, but not every symmetric signal is even. To be even, it must be symmetric about time 0. The system’s signal is periodic, but symmetric to 0° has no effect on the results to be processed.

Zero phase filter methods are as follows [23]. First, the digital signal of input is x(k), and x(k) passes through a bandpass filter. Second, the filtering result $y_{1}\left(k\right)$ is reversed to get the signal $y_{2}\left(k\right)$. Thirdly, $y_{2}\left(k\right)$ passes through a reverse filter. Finally, an accurate signal y(k) which has no phase distortion is output. The filtering algorithms are as follows.
(3)$$y_{1}\left(k\right)=x\left(k\right)*h\left(k\right)$$
(4)$$y_{2}\left(k\right)=y\left(K-1-k\right)$$
(5)$$y_{3}\left(k\right)=y_{2}\left(k\right)*h\left(k\right)$$
(6)$$y\left(k\right)=y_{3}\left(K-1-k\right)$$
where k is the present system state, K is the system total sequence number, and h(k) is the unit impulse signal.

The expressions of Fourier-transform in frequency domain are as follows.

(7)$$Y_{1}\left(e^{j\omega }\right)=X\left(e^{j\omega }\right)H\left(e^{j\omega }\right)$$

(8)$$Y_{2}\left(e^{j\omega }\right)=e^{-j\omega \left(K-1\right)}Y_{1}\left(e^{-j\omega }\right)$$

(9)$$Y_{3}\left(e^{j\omega }\right)=Y_{3}\left(e^{j\omega }\right)H\left(e^{j\omega }\right)$$

(10)$$Y\left(e^{j\omega }\right)=e^{-j\omega \left(K-1\right)}Y_{3}\left(e^{-j\omega }\right)$$

After the arrangement of Equations (7)–(10), it can be obtained that
(11)$$Y\left(e^{j\omega }\right)=X\left(e^{j\omega }\right){\left|H\left(e^{j\omega }\right)\right|}^{2}$$

All in all, there is only an amplitude gain relationship between filter output $Y\left(e^{j\omega }\right)$ and input $X\left(e^{j\omega }\right)$, and there is no phase shift in the full frequency. In theory, zero phase bandpass filter can realize no phase distortion filtering. Besides, signals need to be reversed in time domain during filtering. Therefore, we extract the phase using the Hilbert transform without any noise [24].

### 3.3. Software Subdivision Method

#### 3.3.1. Analog Pulse Counter Method

The APC method to generate the encoder output is through calculating the arctangent value of the signal, and then processing the analog signal. First, the phase angle signal arc tangent is computed; then the arc tangent value is used to simulate the pulse count. Finally, all of the pulse’s corresponding angles are superimposed to generate the output encoder rotation angle. The flowchart of the algorithm is shown in Figure 3; each of the steps is described below.

Arc tangent operation. According to the zero phase bandpass filter output result (phases A and B), the arctangent of each phase angle signal is calculated.

(12)$$T\left(n\right)=\arctan\frac{U_{A}}{U_{B}}$$

In Equation (12), $U_{A}$ and $U_{B}$ are signal values of A and B and n is the corresponding point.

Tangent value overflow pretreatment. Because the tangent value of the signal neighbor at 90° approaches infinity, the overflow processing to the coded signal tangent value is going to overflow. Therefore, the overflowing tangent values are set to the overflowing threshold values. For example, the overflowing threshold values are ±1000 in this paper. Then, the peak values of tangent are ±1000.

Zero-crossing jitter elimination pretreatment. An SCE instrument error causes the sin-cos signal to exhibit the jitter phenomenon. A zero phase bandpass filter signal can still exist within smooth, vertical vibrations. When the judgment is abnormal fluctuation, the value T(n) is pulled to a low level, which can eliminate the noise caused by the cut signal zero-crossing jitters.

Analog pulse counting operation. The subdivision ratio is R, and the Line is L. In the arctangent of the phase angle signal, the analog signal level changes at each pulse width. This indicates a complete pulse signal; the pulse number M is retained in the accumulator, where the pulse width resolution is

(13)$$W=\frac{360{}^{\circ}}{L}/R=\frac{360{}^{\circ}}{R\times L}$$

Rotation angle output. The total rotation angle of the encoder is calculated according to the accumulator count value and the subdivision ratio:(14)$$\theta =M\cdot W=M\times \frac{360{}^{\circ}}{R\times L}$$

#### 3.3.2. Arc Tangent Subdivision Method

The ATS method generates the encoder output rotational angle by calculating the arctangent of the signal. First, the arctangent of the phase angle is calculated. Then, the starting phase angle, the total number of the half-period, and the signal endpoint phase angle of the signal are calculated. Finally, the encoder rotational angle is output. The algorithm flowchart is shown in Figure 4; each of the steps is described below.

Arc tangent operation and preprocessing. According to the output result of the zero phase bandpass filter (phases A and B), the arctangent of the phase angle signal of each point is calculated. The tangent value overflow preprocessing and zero-crossing vibration preprocessing is the same as Equation (12).

Signal starting phase angle calculation. Because the signal does not always to start from half a period (0° or 180°), it is necessary to calculate the nonperiodic part of the phase angle at the starting point of the signal. The physical line number of the encoder is lines; the phase angle signal half-cycle starting point is $n_{0}$; and the signal starts nonperiodic phase angle

(15)$$\theta_{0}=\frac{90{}^{\circ}-T\left(n_{0}\right)}{180{}^{\circ}}\times \frac{360{}^{\circ}}{2\times L}$$

Semicycle count endpoint detection and counting. The arc tangent of the phase angle signal is tested for the zero crossing in 90°~−90°. The arctangent signal zero crossing times, which is N, is determined by the accumulator at 90°~−90°. Then, the total phase angle of half-cycle is

(16)$$\theta_{1}=\left(N-1\right)\times \frac{360{}^{\circ}}{2\times L}$$

Signal endpoint phase angle calculation. Assume the endpoint phase angle of the signal is $\theta_{2}$ and the endpoint phase angle of half-period signal is $n_{t}$. Similar to the starting phase angle computation, the endpoint nonperiodic phase angle of the signal is

(17)$$\theta_{2}=\frac{T\left(n_{t}\right)+90{}^{\circ}}{180{}^{\circ}}\times \frac{360{}^{\circ}}{2\times L}$$

Rotation angle output. Lastly, the total rotation angle of the encoder is calculated.

(18)$$\theta =\theta_{0}+\theta_{1}+\theta_{2}$$

## 4. Case Study Process and Simulation Verification

### 4.1. Simulation of the Analog Pulse Counter

In order to verify the validity and accuracy of the software subdivision methods designed in this paper, a simulation experiment is designed. The simulation system is developed using MATLAB; the parameter values correspond to the actual hardware parameters; the proposed method is realized in the system. The physical line numbers of the SCE test system is L=20000 lines, speed is 200 rpm, and the acquisition speed of the high-speed digitizer is 40 MS/s. The simulation system achieves a high-speed signal collection; the digital phase A and phase B are acquired with a 90° phase difference. The phase A is taken as an example, as shown in Figure 5.

The sin-cos signal of phases A and B (Figure 5) is substituted into the zero phase bandpass filtering algorithm. The algorithm iterates, resulting in a denoised digital signal, with a two phase angle with a 90° difference. The phase A is taken as an example, as shown in Figure 6.

Tangent value overflow processing. The encoding signal tangent value overflow threshold is set to ±1000 and the encoded signal overflow tangent value unified to overflow threshold is set to ±1000 (in the sin signal to judge positive and negative), as shown in Figure 7. The overflow processing value after the absolute tangent is taken is less than or equal to 1000.

The arctangent calculation. According to the zero phase bandpass filter output (phase A and phase B) in Figure 6, the phase angle signal of each point is calculated. This is shown in Figure 8.

Analog pulse counting operation. Selecting the subdivision ratio R as 36, then the analog pulse width resolution is

(19)$$W=\;\frac{360{}^{\circ}}{R\times L}\;=\;0.0005{}^{\circ}$$

The drop along the trigger threshold is set to a 4900 limit to eliminate the zero-crossing jitter elimination phenomenon. The phase angle signal arctangent responds to every pulse width to change the signal level and identify a complete pulse signal. The accumulator is used to obtain the pulse number M as 599. The converted pulse level is shown in Figure 9.

Rotation angle output. Finally, the total rotation angle of the encoder is calculated according to the accumulator count value and the subdivision ratio:(20)$$\theta^{\prime }=M\cdot W=0.2995{}^{\circ}$$

### 4.2. Simulation of the Arc Tangent Subdivision

Based on the arctangent signal in Figure 8, the ATS method is simulated.

Signal starting phase angle calculation. Since the signal does not necessarily start from the half cycle (θ=0° or 180°), it is necessary to calculate the nonperiodic part of the phase angle at the starting point of the signal. The number of physical lines of the known encoder is L=20,000 lines, and the starting point of the signal starting point $n_{0}=1$ is determined by measuring the beginning of the phase angle signal.

(21)$$\theta_{0}=\frac{90{}^{\circ}-T\left(n_{0}\right)}{180{}^{\circ}}\times \frac{360{}^{\circ}}{2\times L}=0.00225{}^{\circ}$$

Semiperiodic interception and counting. The phase angle signal T(n) zero-crossing is tested in the range of 90° to −90°. A total of N=34 times for the zero-crossing in the signal 90°–−90° is obtained by the accumulator. The total angle of the semiperiodic phase angle is

(22)$$\theta_{1}=\left(N-1\right)\times \frac{360{}^{\circ}}{2\times L}=0.297{}^{\circ}$$

Signal end point phase angle calculation. Similar to the starting phase angle, the signal end point phase angle is

(23)$$\theta_{2}=\frac{T\left(n\right)+90{}^{\circ}}{180{}^{\circ}}\times \frac{360{}^{\circ}}{2\times L}\approx 0.00067{}^{\circ}$$

Rotation angle output. Lastly, the total rotation angle of the encoder is calculated.

(24)$$\theta =\theta_{0}+\theta_{1}+\theta_{2}=0.2999{}^{\circ}$$

### 4.3. Comparison and Analysis of Simulation Results

Assume the rotational speed is r and the rotation time is t. Then, the actual theoretical angle of the encoder rotation is

(25)$$\theta^{\prime \prime }=\frac{r}{60}\cdot t\cdot 360{}^{\circ}=\frac{200}{60}\times 2.5\times 10^{-4}\times 360{}^{\circ}=0.3{}^{\circ}$$

In the simulation results, the total rotation angles of the encoder obtained by ATS and APC methods are compared with the theoretical angle. The results show that both methods can achieve a more accurate angle. The maximum error of the APC method is the corresponding angle of the pulse width. However, the physical calculated precision of the encoder can be achieved by the ATS method.

The ATS method is suitable for use in the industrial computer environment, and its accuracy is higher. The APC method has a lower precision than that of the ATS method. However, the APC method is easier to run stably in embedded systems or in systems with stringent real-time requirement in practical engineering applications.

## 5. Experiment Validation

### 5.1. Experiment Contrast before and after Zero Phase Bandpass Filter

The validation of the software subdivision has been performed in a high-resolution SCE test system of the precision reducer test platform. The precision reducer test platform is shown in Figure 10. The precision reducer test platform is a device for testing the comprehensive performance of reducer, which is used to verify the rationality of reducer design. The SCE test system is a high precision acquisition device for reducer signal on the precision reducer test platform.

The physical line number of the SCE test system is 20,000 lines; the speed is 300 rpm; and the acquisition speed of the high-speed digitizer is 40 MS/s. Through the SCE test system, taking the ATS method as an example, two experiments with and without the zero phase bandpass filter are carried out. The experimental data are plotted by MATLAB to obtain the sin-cos signal waveform contrast diagram before and after zero phase bandpass filter. The experiments use the sin signal as an example, as shown in Figure 11. The error between the output angle and the encoder theory rotation angle are calculated (with and without the filter). The angle deviation graph is drawn, as shown in Figure 12.

First, the influence of the zero phase bandpass filter on the calculation of the encoder rotation angle is analyzed for the experimental results, before and after the zero phase bandpass filter. Through the waveform comparison results of Figure 11, it can be seen that the noise near the peak value of the filter is larger, which becomes one of the error sources in the angle calculation of the subsequent encoder. The filtered waveform tends to be stable and smooth; the denoising effect is demonstrated, and it effectively improves the angle error brought by the waveform. The results of the encoder angle comparison, presented in Figure 12, are as follows.

(1)The peak of the deviation after the filtering is smaller than before the filter and the overall error decreases.(2)The standard deviation of the filter deviation is smaller than that before the filtering and the accidental error before the filtering is eliminated.(3)The negative overall tendency of the filtered deviation is due to the phase offset caused by the zero phase bandpass filtering algorithm. However, this offset has no effect on the relative rotation angle of the calculation.

### 5.2. Experimental Comparison of the Software Subdivision Methods

In the above test experiment of the high-precision SCE, the ATS and the APC methods are used. Compared with the actual theoretical angle, the deviation graph of the proposed software subdivision method is presented in Figure 13.

Comparing the experimental results of the ATS method and the analogy pulse counting method, the influence of the two software subdivision methods on the accuracy of the calculation of the encoder rotation angle is analyzed. The results of the encoder angle comparison, presented in Figure 13, as follows.

(1)The deviation of the ATS method is one order of magnitude lower than that of the APC method, and the accuracy is relatively higher.(2)The output from the APC method is a digital quantity. This value is affected by the system resolution and generates a pulse system error.

In this paper, a novel subdivision system of the high-precision SCE based on a zero phase bandpass filter is designed, which can effectively solve the noise problem caused by the hardware and the environment. Two methods are explored: ATS and APC. The ATS method offers high precision and the APC provides a relatively low precision, but offers a fast, stable option for practical, real-time applications.

## 6. Conclusions

In order to solve the problems of large signal noise and low subdivision accuracy in the high-speed sampling of encoder signals, this paper designs a software subdivision system based on zero phase bandpass filter for high-precision SCE. The conclusions can be summarized as follows.

(1)According to the acquisition characteristics of high speed and high-precision SCE, a novel software subdivision system is designed. The method’s steps are straightforward, providing a solution that is fast, accurate, and easy to program.(2)Through the zero phase bandpass filter algorithm denoising, the filtered waveform tends to be stable and smooth, realizes the denoising effect, and effectively improves the angle error caused by the noise.(3)The ATS method can be directly used in the subdivision on the sin-cos signal, which is more practical in an offline situation and has higher accuracy.(4)By using the APC method, the converted pulse data can be imported into the embedded system to run. This provides better compatibility in comparison to the tangent subdivision method; it is more suitable for applications with stringent real-time requirements.

Through the simulation analysis and the experimental verification, it is demonstrated that the proposed approach can be applied to the signal acquisition and the software segmentation of a high-precision SCE. The software subdivision method has the characteristics of high precision, easy integration, and low cost.

In future, more research work should continually focus on the improvement for high-precision subdivision system. The research can also be used in the field of fault diagnosis and condition monitoring, such as reducer fault diagnosis or condition monitoring of railway vehicles and track. However, in most of complex systems, the monitoring data involves multiple types of parameters. Thus, a comprehensive method which can both handle the continuous and discrete parameters is essential for actual industrial applications. The presented system and its application to the envelope demodulation of reducer shock signal based on peak retention and downsampling might be developed by other groups of researchers which are developing creative methods of research, e.g., Kostrzewski [25].

## Figures and Tables

**Figure 1 sensors-19-03041-f001:**
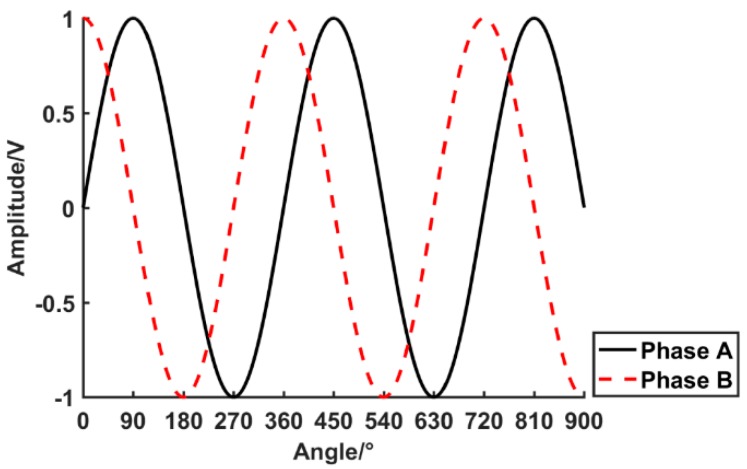
Voltage signals of phases A and B.

**Figure 2 sensors-19-03041-f002:**
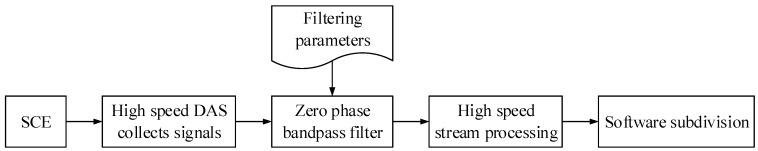
Flowchart of algorithm.

**Figure 3 sensors-19-03041-f003:**
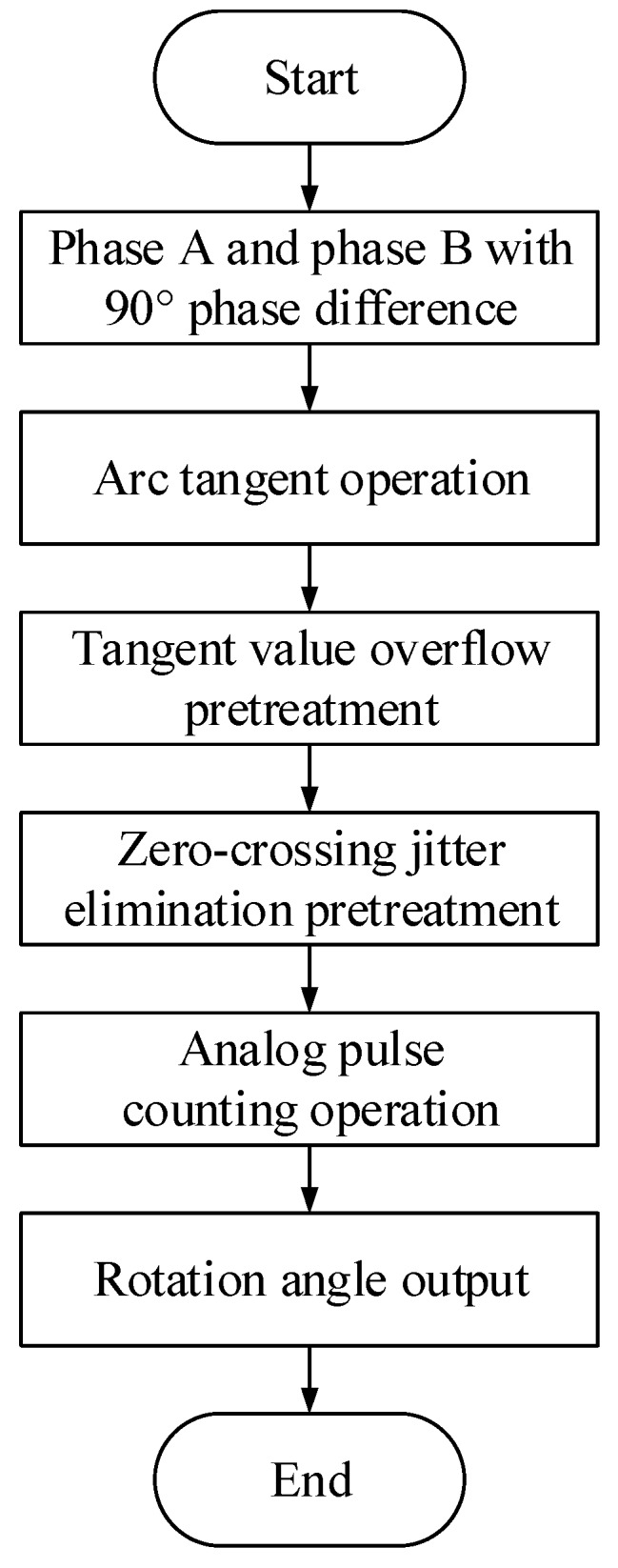
Technical process of the Analog Pulse Counter (APC) method.

**Figure 4 sensors-19-03041-f004:**
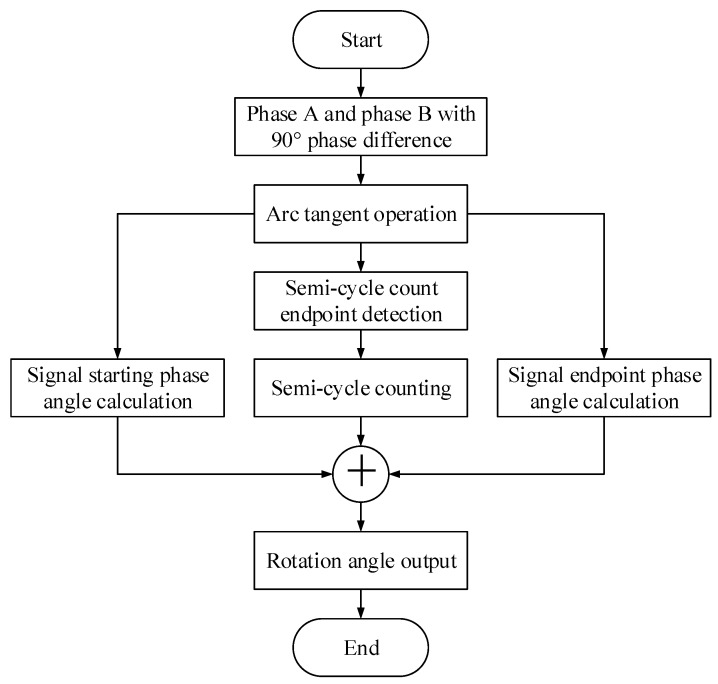
Technical process of the Arc Tangent Subdivision (ATS) method.

**Figure 5 sensors-19-03041-f005:**
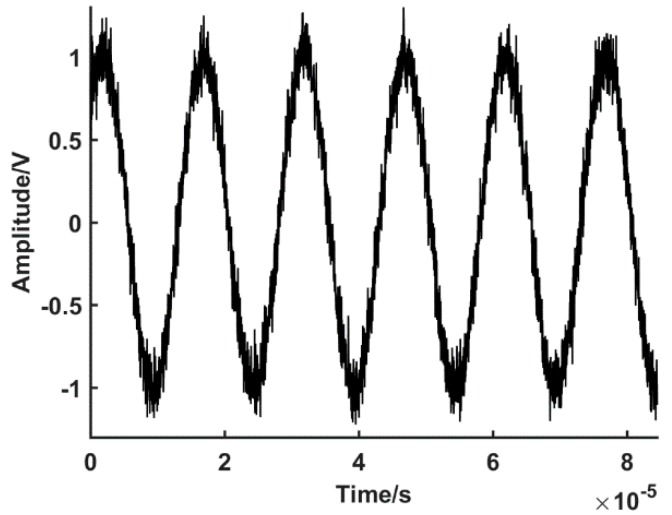
Sinusoidal signal acquired by high-speed data acquisition system (DAS).

**Figure 6 sensors-19-03041-f006:**
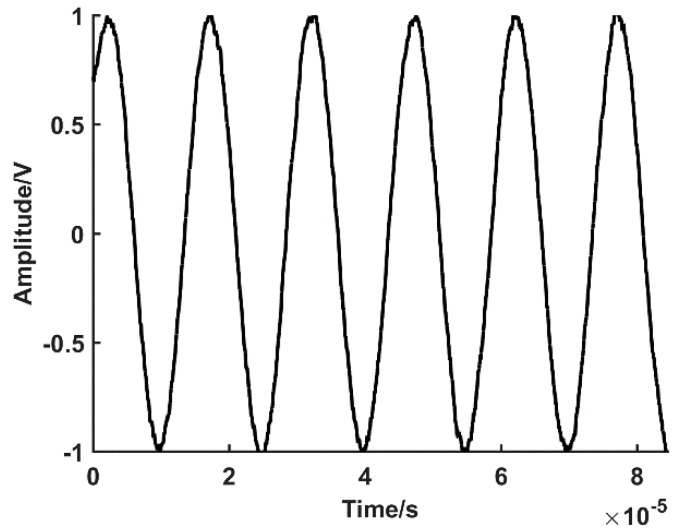
Sinusoidal signal with noise elimination by zero phase bandpass filter.

**Figure 7 sensors-19-03041-f007:**
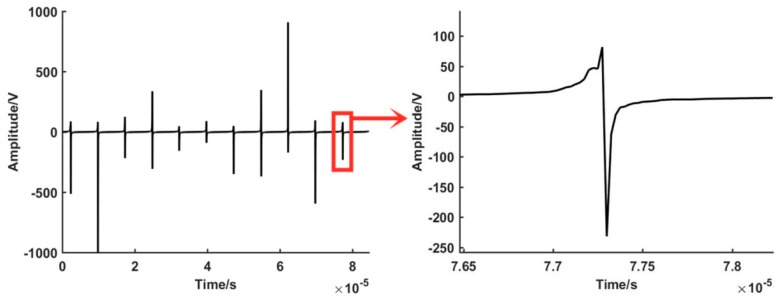
Overflowing handling of the tangent value of encoder output.

**Figure 8 sensors-19-03041-f008:**
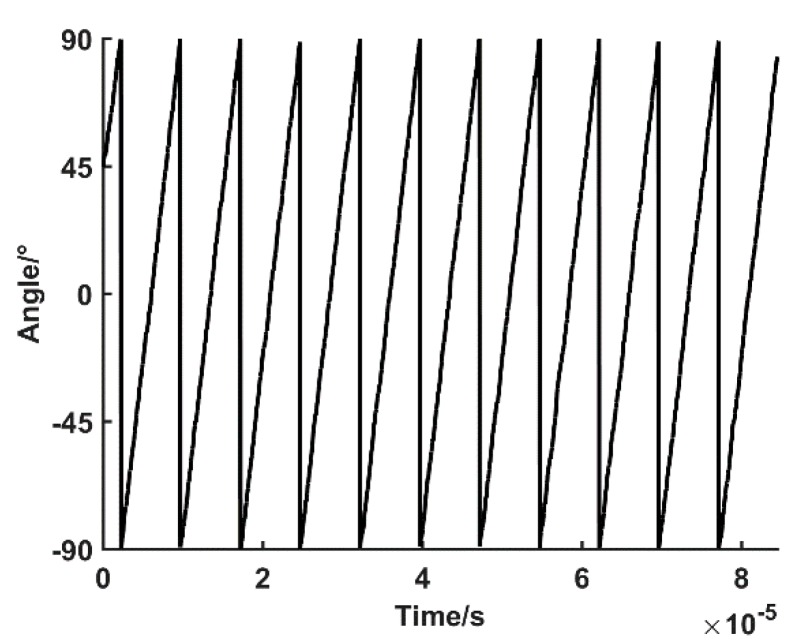
Arc tangent results of two channels signal.

**Figure 9 sensors-19-03041-f009:**
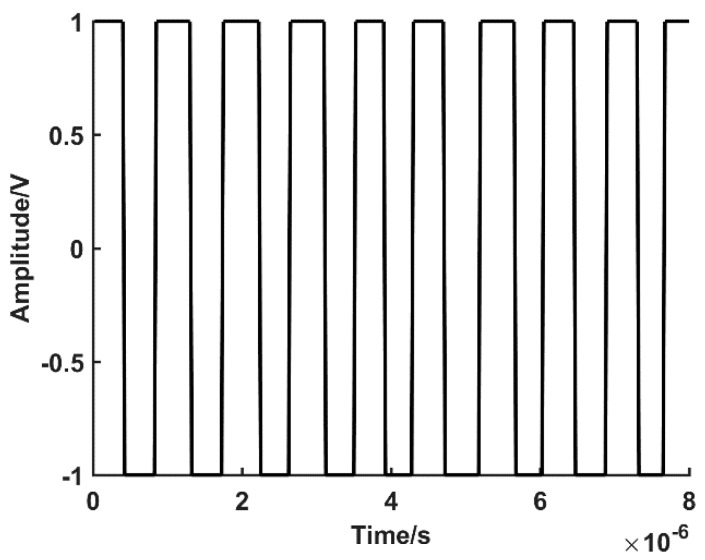
The transformed result of analog pulse signal.

**Figure 10 sensors-19-03041-f010:**
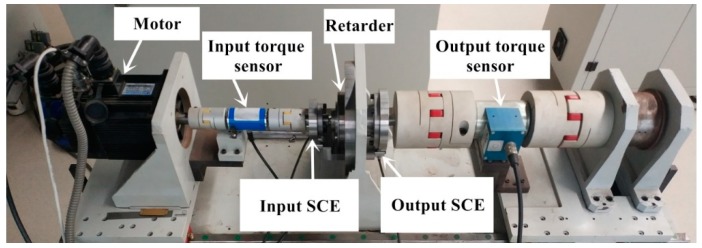
The test bench of precision speed reducer.

**Figure 11 sensors-19-03041-f011:**
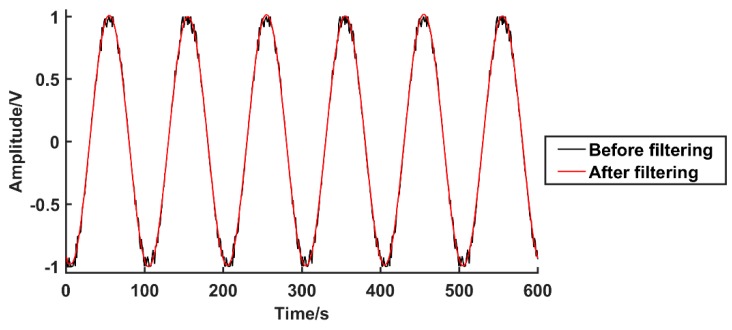
Sinusoidal signals before and after zero phase bandpass filter.

**Figure 12 sensors-19-03041-f012:**
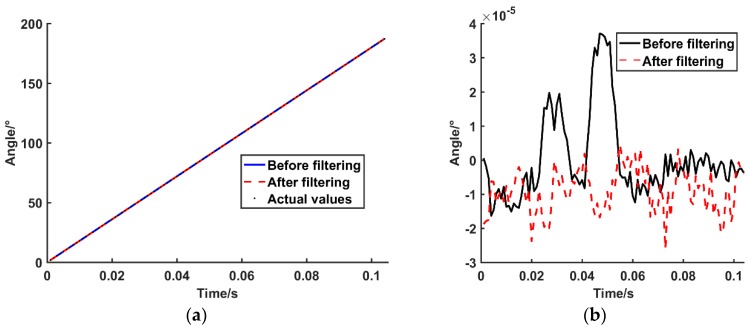
Result comparison before and after filtering. (**a**) The encoder rotation angles obtained by ATS method before and after filtering. (**b**) Angular deviation between calculated results and actual values before and after filtering.

**Figure 13 sensors-19-03041-f013:**
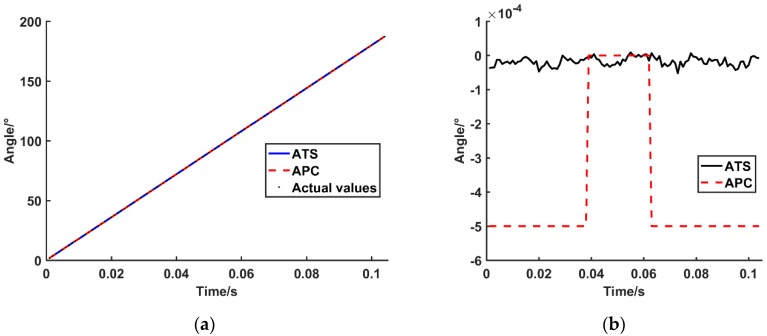
Angle comparison of different subdivision methods. (**a**) The encoder rotation angle obtained by two subdivision methods. (**b**) Angular deviation between calculated results and actual values with different subdivision methods.

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
