# Peer review of "Research on Subdivision System of Sin-Cos Encoder Based on Zero Phase Bandpass Filter"

_sensors, 2019, doi:10.3390/s19143041_

Round 1
Reviewer 1 Report
The paper is improved now, however it still needs some changes given below.
Parallel streams processing technologies that the high-speed streams of large file data import disks are adopted to achieve this - is this the latest technology? Is it up-to-date?
Description parameter „t” in eq. (1),(2) is missing. We cannot assume that everybody might know its meaning.
Change „Equation (1) and (2) (…) the input amplitude….”
into „Equations (1) and (2) (…) the input amplitude is…”. „Besides, Signals” into „Besides, signals”
What made you changed Kalman filter into Zero phase bandpass filter ? Zero phase bandpass filter is symmetric to 0° - please, explain it for your research.
In many spots spaces are missing and letters should be big - please, verify the paper again.
Please, describe the equipment used for research.
Future research plans might be given in the conclusion section as well. How can high precision subdivision system for high speed encoders be improved? Could design thinking method be used by the research team? https://doi.org/10.1007/978-3-319-95204-8_16
Author Response
Response to Reviewers’ Comments for Reviewer #1
Sensors- 540402: Research on Subdivision System of Sin-Cos Encoder Based on Zero Phase Bandpass Filter
Haoning Zhao, Jiazhong Xu*, Haibin Zhang*, Zhen Liu and Shi Dong
Reviewer’s Comment Point 1:
The paper is improved now, however it still needs some changes given below.
Authors’ Response 1:
The authors are grateful to the reviewer for the positive comments on the contributions of our paper, and the affirmation of our work. The referee’s concern and suggestion are also quite valuable and stimulative to our research and all the comments and suggestions have been carefully considered. The following summarizes our revisions and explanations itemized according to the reviewer’s comments (Note: unless otherwise specified, all the page numbers cited below are for the revised manuscript). All locations of the revisions to the manuscript are highlighted with the YELLOW background color and detailed revision contents are characterized with the red font.
Reviewer’s Comment Point 2:
Parallel streams processing technologies that the high-speed streams of large file data import disks are adopted to achieve this - is this the latest technology? Is it up-to-date?
Authors’ Response 2:
Thanks to the reviewer for the meritorious suggestion. The reviewer’s question is meritorious and we have also considered it comprehensively. Parallel streams processing technology is not the latest technology. But it has been widely used recently. Its processing speed and capability are determined by the running speed and processing capacity of the computer. In recent years, the remarkable increase in the speed of computer operation has made this technology more and more powerful. In the large amount of data processing of the subdivision system, it significantly improves the speed of operation and plays an indispensable role.
Reviewer’s Comment Point 3:
Description parameter “t” in eq. (1), (2) is missing. We cannot assume that everybody might know its meaning.
Authors’ Response 3:
The authors would like to thank the reviewer for giving us a chance to revise the paper, and also thank the reviewers for giving us constructive suggestions which would help us in depth to improve the quality of the paper. Sorry about the mistake and thanks to the reviewer’s tip. It is really true as Reviewer suggested that description parameter “t” in eq. (1), (2) is missing. So the authors have added the description parameter “t” in the text as below:
“t is time.” (Lines 131, Page 4 in the revised manuscript)
Reviewer’s Comment Point 4:
Change “Equation (1) and (2) (…) the input amplitude….” into “Equations (1) and (2) (…) the input amplitude is…”. “Besides, Signals” into “Besides, signals”
Authors’ Response 4:
Sorry about the mistake and thanks to the reviewer’s tip. As Reviewer suggested that “Equation (1) and (2) (…) the input amplitude….” is changed into form of “Equations (1) and (2) (…) the input amplitude is…”. (Lines 136, Page 4 in the revised manuscript)
“Besides, Signals” is replace by “Besides, signals”. (Lines 195, Page 6 in the revised manuscript)
Reviewer’s Comment Point 5:
What made you changed Kalman filter into Zero phase bandpass filter? Zero phase bandpass filter is symmetric to 0° - please, explain it for your research.
Authors’ Response 5:
Thank you very much for the comments. The reviewer’s concern is quite significant and this is also considered by us. As the previous reviewer said, Estimation of the phase of a sine signal with known and more importantly constant frequency is a very simple problem. Instantaneous phase is a result of a simple Hilbert transform of the signal. If the Kalman filter has one state and that is the phase, there is no need for any further processing. The phase is just a result of the Kalman filter. The zero crossings can be estimated and hence the speed. So we can simply apply zero phase bandpass filter to the signal and then extract the phase using the Hilbert transform without any noise. Usually, the signal will have phase shift after filtering in time domain. Zero phase filtering is a digital filtering technology to solve this problem. The phase response of zero phase filter system function is zero. Zero phase filtering is an excellent solution to the signal processing process which needs to keep the signal phase after filtering. Note that every even signal is symmetric, but not every symmetric signal is even. To be even, it must be symmetric about time 0. The system’s signal is periodic, but symmetric to 0° has no effect on the results to be processed.
Reviewer’s Comment Point 6:
In many spots spaces are missing and letters should be big - please, verify the paper again.
Authors’ Response 6:
Reviewer’s suggestion is greatly appreciated. We are grateful to the reviewer about the careful work, and we have made correction according to the Reviewer’s comments. We added spots spaces and changed the font size of the letters.
Reviewer’s Comment Point 7:
Please, describe the equipment used for research.
Authors’ Response 7:
The authors would like to thank the reviewer for the constructive review. To clarify this case, the authors have added the section 5.1 in the text as below:
“The validation of the software subdivision has been performed in a high-resolution SCE test system of the precision reducer test platform. The precision reducer test platform is shown in Figure 10. The precision reducer test platform is a device for testing the comprehensive performance of reducer, which is used to verify the rationality of reducer design. The SCE test system is a high precision acquisition device for reducer signal on the precision reducer test platform.” (Lines 323-328, Page 13 in the revised manuscript)
Reviewer’s Comment Point 8:
Future research plans might be given in the conclusion section as well. How can high precision subdivision system for high speed encoders be improved? Could design thinking method be used by the research team? https://doi.org/10.1007/978-3-319-95204-8_16
Authors’ Response 8:
Thanks to the reviewer for the meritorious suggestion. The reviewer’s question is meritorious and we have also considered it comprehensively. According to the reviewer’s suggestion, we have added the future research plans in the conclusion section. The revisions are listed in detail as follows:
“In future, more research work should continually focus on the improvement for high precision subdivision system. The research can also be used in the field of fault diagnosis and condition monitoring, such as reducer fault diagnosis or condition monitoring of railway vehicles and track. However, in most of complex systems, the monitoring data involves multiple types of parameters. Thus, a comprehensive method which can both handle the continuous and discrete parameters is essential for actual industrial applications. We will improve the system and apply it to the envelope demodulation of reducer shock signal based on peak retention and downsampling, and the method be used by the research team [25].” (Lines 399-406, Page 17 in the revised manuscript)
“25. Kostrzewski, M. One Design Issue – Many Solutions. Different Perspectives of Design Thinking – Case Study[J]. Communications in Computer and Information Science. 2018, 877, 179-190.” (Lines 471-472, Page 18 in the revised manuscript)
Finally, the authors would like to thank the reviewer again for the constructive comments and hope that the revisions are some satisfactory. We have updated the acknowledgments to express sincere appreciations to the reviewer for the valuable comments to further improve the paper.
Yours sincerely,
Haoning Zhao, Jiazhong Xu, Haibin Zhang, Zhen Liu and Shi Dong

Reviewer 2 Report
The paper presents methods for software subdivision from Sin-Cos Encoder based zero phase bandpass filter. The work is significant for developing such a reliable and yet cost-effective subdivision system. I consider the manuscript an interesting application of signal theory to subdivision system. The authors provided a supplementary file that confirms proof reading of the manuscript. The manuscript is well written, and the topic is interesting. But, the following problem should be addressed before it is accepted.
1. Line 84-85: The description of this sentence is incorrect. Now the system capability of DSP and FPGA is very strong. There is no such situation as the described. Considering that the description and the main content of the article are not relevant, it is suggested to delete these relevant content.
2. Some engineering application environments can be added to the background.
3. Line 106-108: What are the parallel streams processing technologies? Please confirm its relationship with the main contents of the paper.
4. The author should explain all the parameters mentioned in the paper.
5. Line 255-256: What is the source of noise in the sine signal in Figure 5? Is it artificial random noise? Please explain the effect of noise on validating the filter.
6. Line 370-389: It is suggested that the conclusions describe the innovative points of the paper and does not focus on describing well-known methods and techniques.
7. Please explain why you use zero phase bandpass filter in subdivision system. What are the Advantages?
Author Response
Response to Reviewers’ Comments for Reviewer #2
Sensors- 540402: Research on Subdivision System of Sin-Cos Encoder Based on Zero Phase Bandpass Filter
Haoning Zhao, Jiazhong Xu*, Haibin Zhang*, Zhen Liu and Shi Dong
Reviewer’s Comment Point 1:
The paper presents methods for software subdivision from Sin-Cos Encoder based zero phase bandpass filter. The work is significant for developing such a reliable and yet cost-effective subdivision system. I consider the manuscript an interesting application of signal theory to subdivision system. The authors provided a supplementary file that confirms proof reading of the manuscript. The manuscript is well written, and the topic is interesting. But, the following problem should be addressed before it is accepted.
Authors’ Response 1:
The authors are such grateful to the reviewer for the positive comments on the contributions of our paper, and the affirmation of our work. The referee’s following concern and suggestions are also quite valuable and stimulative to our research and all the comments have been carefully considered. The following parts summarize our revisions and explanations itemized according to the reviewer’s comments (Note: unless otherwise specified, all the page numbers cited below are for the revised manuscript). All locations of the revisions to the manuscript are highlighted with the YELLOW background color and detailed revision contents are characterized with the red font.
Reviewer’s Comment Point 2:
Line 84-85: The description of this sentence is incorrect. Now the system capability of DSP and FPGA is very strong. There is no such situation as the described. Considering that the description and the main content of the article are not relevant, it is suggested to delete these relevant content.
Authors’ Response 2:
We would like to thank the reviewer for giving us a chance to revise the paper, and also thank the reviewers for giving us constructive suggestions which would help us both in English and in depth to improve the quality of the paper. Sorry about the mistake and thanks to the reviewer’s tip. It is really true as Reviewer suggested that the description and the main content of the article are not relevant. So we have removed it in the revised manuscript according to the reviewer’s comment. The following parts in blue font are removed:
“The operation speed and pulse processing ability of these computers are more advantageous than the DSP and FPGA controllers.” (Lines 89-90, Page 2 in the revised manuscript)
Reviewer’s Comment Point 3:
Some engineering application environments can be added to the background.
Authors’ Response 3:
Thank you very much for the comments on the paper and the research significance of the study. We have re-written this part according to the Reviewer’s suggestion, and we have added to the discussion of the background such as follows:
“High precision test data is an important guarantee of the reducer testing system. The signal acquisition system of the reducer is an important part of the reducer testing. It is closely related to the accuracy of the test data of the reducer [3].” (Lines 46-49, Page 2 in the revised manuscript)
Reviewer’s Comment Point 4:
Line 106-108: What are the parallel streams processing technologies? Please confirm its relationship with the main contents of the paper.
Authors’ Response 4:
Thanks to the reviewer for the meritorious suggestion. The reviewer’s question is meritorious and we have also considered it comprehensively. In order to express the point more clearly according to the reviewer’s comments, we add this part as a discussion as follows:
“Parallel streams processing technologies that the high-speed streams of large file data import disks are adopted to achieve this.” (Lines 111-113, Page 3 in the revised manuscript)
Reviewer’s Comment Point 5:
The author should explain all the parameters mentioned in the paper.
Authors’ Response 5:
Many thanks to the reviewer about the valuable suggestion. The reviewer’s concern is quite significant and we have defined all parameters in detail.
Reviewer’s Comment Point 6:
Line 255-256: What is the source of noise in the sine signal in Figure 5? Is it artificial random noise? Please explain the effect of noise on validating the filter.
Authors’ Response 6:
Thanks to the reviewer for the meritorious suggestion. The reviewer’s concern is also what we have considered. As the reviewer said, the noise of the sine signal in Figure 5 is artificial random noise. Because the noise is unavoidable in practical engineering, the authors add random noise by the Matlab program in the simulation verification. The results can be seen that the noise near the peak value of the filter is larger, which becomes one of the error sources in the angle calculation of the subsequent encoder. The filtered waveform tends to be stable and smooth; the denoising effect is demonstrated, and it effectively improves the angle error brought by the waveform. The peak of the deviation after the filtering is smaller than before the filter, and the overall error decreases.The standard deviation of the filter deviation is smaller than that before the filtering, and the accidental error before the filtering is eliminated.
Reviewer’s Comment Point 7:
Line 370-389: It is suggested that the conclusions describe the innovative points of the paper and does not focus on describing well-known methods and techniques.
Authors’ Response 7:
Thanks to the reviewer for the meritorious suggestion. The reviewer’s question is meritorious and we have also considered it comprehensively. In order to express the point more clearly according to the reviewer’s comments, we add this part as a discussion as follows:
“In order to solve the problems of large signal noise and low subdivision accuracy in the high-speed sampling of encoder signals, this paper designs a software subdivision system based on a zero phase bandpass filter for high precision SCE. The conclusions can be summarized as follows.
(1) According to the acquisition characteristics of high speed and high precision SCE, a novel software subdivision system is designed. The method’s steps are straightforward, providing a solution that is fast, accurate, and easy to program.
(2) Through the zero phase bandpass filter algorithm denoising, the filtered waveform tends to be stable and smooth, realizes the denoising effect, and effectively improves the angle error caused by the noise.
(3) The ATS method can be directly used in the subdivision on the sin-cos signal, which is more practical in an offline situation and has higher accuracy.
(4) By using the APC method, the converted pulse data can be imported into the embedded system to run. This provides better compatibility in comparison to the tangent subdivision method; it is more suitable for applications with stringent real-time requirements.
Through the simulation analysis and the experimental verification, it is demonstrated that the proposed approach can be applied to the signal acquisition and the software segmentation of a high precision SCE. The software subdivision method has the characteristics of high precision, easy integration, and low cost.
In future, more research work should continually focus on the improvement for high precision subdivision system. The research can also be used in the field of fault diagnosis and condition monitoring, such as reducer fault diagnosis or condition monitoring of railway vehicles and track. However, in most of complex systems, the monitoring data involves multiple types of parameters. Thus, a comprehensive method which can both handle the continuous and discrete parameters is essential for actual industrial applications. We will improve the system and apply it to the envelope demodulation of reducer shock signal based on peak retention and downsampling, and the method be used by the research team [25].” (Lines 380-406, Page 16-17 in the revised manuscript)
Reviewer’s Comment Point 8:
Please explain why you use zero phase bandpass filter in subdivision system. What are the Advantages?
Authors’ Response 8:
Thank you very much for the comments. As the reviewer said, the authors use zero phase bandpass filter in subdivision system. Usually, the signal will have phase shift after filtering in time domain. Zero phase filtering is a digital filtering technology to solve this problem. The phase response of zero phase filter system function is zero. Zero phase filtering is an excellent solution to the signal processing process which needs to keep the signal phase after filtering.
Finally, thank you again for your efforts on our work and hope that we have answered your question and revised the manuscript according to your comments satisfyingly.
Yours sincerely,
Haoning Zhao, Jiazhong Xu, Haibin Zhang, Zhen Liu and Shi Dong

Round 2
Reviewer 1 Report
I think that "Authors’ Response 5" might be partly written in the manuscript. I also suggest Authors to change: "We will improve the system and apply it to the envelope demodulation of reducer shock signal based on peak retention and downsampling, and the method be used by the research team" into "The presented system and its application to the envelope demodulation of reducer shock signal based on peak retention and downsampling might be developed by other groups of researchers which are developing creative methods of research, e.g. [25]"
Author Response
Response to Reviewers’ Comments for Reviewer #1 (Round 2)
Sensors- 540402: Research on Subdivision System of Sin-Cos Encoder Based on Zero Phase Bandpass Filter
Haoning Zhao, Jiazhong Xu*, Haibin Zhang*, Zhen Liu and Shi Dong
Reviewer’s Comment Point 1:
I think that "Authors’ Response 5" might be partly written in the manuscript. I also suggest Authors to change: "We will improve the system and apply it to the envelope demodulation of reducer shock signal based on peak retention and downsampling, and the method be used by the research team" into "The presented system and its application to the envelope demodulation of reducer shock signal based on peak retention and downsampling might be developed by other groups of researchers which are developing creative methods of research, e.g. [25]"
Authors’ Response 1:
The authors are such grateful to the reviewer for the positive comments on the contributions of our paper, and the affirmation of our work. The referee’s following concern and suggestions are also quite valuable and stimulative to our research and all the comments have been carefully considered. The following parts summarize our revisions and explanations itemized according to the reviewer’s comments (Note: unless otherwise specified, all the page numbers cited below are for the revised manuscript). All locations of the revisions to the manuscript are highlighted with the YELLOW background color and detailed revision contents are characterized with the red font.
Thank you very much for the comments on the paper and the research significance of the study. We have revised the manuscript as suggested and re-organized to highlight the key issues all through the paper. In order to cover the reviewer’s suggestion, we have added the following interpretation:
“The zero crossings can be estimated and hence the speed. So we can simply apply zero phase bandpass filter to the signal and then extract the phase using the Hilbert transform without any noise. Usually, the signal will have phase shift after filtering in time domain. Zero phase filtering is a digital filtering technology to solve this problem. The phase response of zero phase filter system function is zero. That is to say, there is no phase deviation after filtering [21]. Zero phase filtering is an excellent solution to the signal processing process which needs to keep the signal phase after filtering [22]. Zero phase filtering is an excellent solution to the signal processing process which needs to keep the signal phase after filtering. Note that every even signal is symmetric, but not every symmetric signal is even. To be even, it must be symmetric about time 0. The system’s signal is periodic, but symmetric to 0° has no effect on the results to be processed.” (Lines 180-189, Page 5 in the revised manuscript)
“We will improve the system and apply it to the envelope demodulation of reducer shock signal based on peak retention and downsampling, and the method be used by the research team [25].” is replace by
“The presented system and its application to the envelope demodulation of reducer shock signal based on peak retention and downsampling might be developed by other groups of researchers which are developing creative methods of research, e.g. [25].” (Lines 409-412, Page 17 in the revised manuscript)
Finally, thank you again for your efforts on our work and hope that we have answered your question and revised the manuscript according to your comments satisfyingly.
Yours sincerely,
Haoning Zhao, Jiazhong Xu, Haibin Zhang, Zhen Liu and Shi Dong

This manuscript is a resubmission of an earlier submission. The following is a list of the peer review reports and author responses from that submission.
Round 1
Reviewer 1 Report
Line 87-88: The word “SCE” is suggested to be given the full name. So do plenty of other titles.
Line 96-100: The texts are suggested to be left out.
The word “MATLAB” is suggested to be used big letters.
Line 241: The figure 7 is suggested to be narrowed.
Line 246,253: The symbol “R” and “M” are suggested to be used italics.
More up-to-date papers are suggested to be referenced.
How is the “encoder rotation angles” obtained in the experiment verification? This should be explained as the different results indicate the effectiveness of the theory.
Please explain why you use two methods in Software subdivision method
Author Response
Response to Reviewers’ Comments for Reviewer #1
Sensors- 492915: Research on Subdivision System of Sin-Cos Encoder Based on Kalman Filtering
Haoning Zhao, Jiazhong Xu*, Haibin Zhang*, Zhen Liu and Shi Dong
The authors are such grateful to the reviewer for the positive comments on the contributions of our paper, and the affirmation of our work. The referee’s following concern and suggestions are also quite valuable and stimulative to our research and all the comments have been carefully considered. The following parts summarize our revisions and explanations itemized according to the reviewer’s comments (Note: unless otherwise specified, all the page numbers cited below are for the revised manuscript). All locations of the revisions to the manuscript are highlighted with the YELLOW background color and detailed revision contents are characterized with the red font.
Reviewer’s Comment Point 1:
Line 87-88: The word “SCE” is suggested to be given the full name. So do plenty of other titles.
Authors’ Response 1:
We are grateful to the reviewer about the careful work. Considering the Reviewer’s suggestion, we have given the full name in titles of chapters and subchapters.
Reviewer’s Comment Point 2:
Line 96-100: The texts are suggested to be left out.
Authors’ Response 2:
We would like to thank the reviewer for giving us a chance to revise the paper, and also thank the reviewers for giving us constructive suggestions which would help us both in English and in depth to improve the quality of the paper. Sorry about the mistake and thanks to the reviewer’s tip. It is really true as Reviewer suggested that this part of content has no meaning to the theme. So we have removed it in the revised manuscript according to the reviewer’s comment. The following parts in blue font are removed:
“A rotary transformer is an absolute magnetoelectric encoder, which is suitable for harsh environments. Although the output is a sine analog signal, its original edge needs to enter a high-frequency excitation signal. This transformer needs a special decoding circuit and a decoding chip; it only outputs a sin-cos period in a week. The subdivision accuracy of a rotary transformer is limited; it cannot reach the high precision of ordinary incremental encoder. Consequently, an SCE has advantages for high-precision and precision test systems [12].” (Lines 96-101, Page 3 in the revised manuscript)
Reviewer’s Comment Point 3:
The word “MATLAB” is suggested to be used big letters.
Authors’ Response 3:
We are grateful to the reviewer about the careful work. The wrong form of “Matlab” is replaced by “MATLAB”.
Reviewer’s Comment Point 4:
Line 241: The figure 7 is suggested to be narrowed.
Authors’ Response 4:
Thank you for the suggestion. As Reviewer suggested that we have narrowed the figure 7. (Lines 251, Page 9 in the revised manuscript)
Reviewer’s Comment Point 5:
Line 246,253: The symbol “R” and “M” are suggested to be used italics.
Authors’ Response 5:
Many thanks to the reviewer about the valuable suggestion, and we have used italics for all variable. Considering the Reviewer’s suggestion, we have used italics for the symbol “R” and “M”.
Reviewer’s Comment Point 6:
More up-to-date papers are suggested to be referenced.
Authors’ Response 6:
Thank you very much for the comments on the paper. We have made correction according to the Reviewer’s comments. As Reviewer suggested that we have referenced more up-to-date papers as follows:
“1. Dogsa, T.; Solar, M.; Jarc, B. Precision Delay Circuit for Analog Quadrature Signals in Sin/Cos Encoders[J]. IEEE Transactions on Instrumentation and Measurement. 2014, 63, 2795-2803.
2. Chudzikiewicz, A.; Bogacz, R.; Kostrzewski, M.; et al. Condition Monitoring of Railway Track Systems by Using Acceleration Signals on Wheelset Axle-Boxes[J]. Transport. 2018, 33, 30-42.
13. Zhao, L.; Cheng, K.; Chen, S.; et al. An approach to investigate moire patterns of a reflective linear encoder with application to accuracy improvement of a machine tool[J]. Proceedings of the Institution of Mechanical Engineers Part B-Journal of Engineering Manufacture. 2019, 233, 927-936.
15. Pablo, M.; Lai, H.; Oppenheim, A. Delta-Ramp Encoder for Amplitude Sampling and Its Interpretation as Time Encoding[J]. IEEE Transactions on Signal Processing. 2019, 67, 2516-2527.
16. Wu, Q.; Shi, F. A backward approach to certain class of transport equations in any dimension based on the Shannon sampling theorem[J]. International Journal of Computer Mathematics. 2017, 94, 1943-1967.” (Lines 375-407, Page 14 in the revised manuscript)
Reviewer’s Comment Point 7:
How is the “encoder rotation angles” obtained in the experiment verification? This should be explained as the different results indicate the effectiveness of the theory.
Authors’ Response 7:
Thanks to the reviewer for the meritorious suggestion. The reviewer’s question is meritorious and we have also considered it comprehensively. The “encoder rotation angles” were obtained in the high-resolution SCE test system by ATS and APC subdivision methods. It is really true as Reviewer suggested that this is explained as the different results indicate the effectiveness of the theory. In the experiment verification, the total rotation angle of the encoder obtained by ATS and APC methods are compared with the theoretical angle. The results show that both methods can achieve a more accurate angle.
Reviewer’s Comment Point 8:
Please explain why you use two methods in Software subdivision method.
Authors’ Response 8:
The reviewer’s concern is quite significant. Because the two methods in Software subdivision method have complementary advantages. The maximum error of the APC method is the corresponding angle of the pulse width. However, the physical calculated precision of the encoder can be achieved by the ATS method. Secondly, the ATS method can be directly used in the subdivision on the sin-cos signal, which is more practical in an offline situation and has higher accuracy. By using the APC method, the converted pulse data can be imported into the embedded system to run. This provides better compatibility in comparison to the tangent subdivision method; it is more suitable for applications with stringent real-time requirements.
Finally, thank you again for your efforts on our work and hope that we have answered your question and revised the manuscript according to your comments satisfyingly.
Yours sincerely,
Haoning Zhao, Jiazhong Xu, Haibin Zhang, Zhen Liu and Shi Dong

Reviewer 2 Report
First paragraph need references in a lot of fragments. So do plenty of other paragraphs.
Authors should avoid acronyms in titles of chapters and subchapters.
Line 69: it should be „It can be observed that…”
Shannon sampling theory is given without any references.
Why do Authors use PHI instead of φ?
Symbol R is covariance once and later is described as subdivision ratio.
Authors could describe verification (and validation) of the model or method, but not a simulation.
Could Authors change „lines” into a symbol?
Matlab is given in big letters once and later on in small letters.
Some parameters are given in straight form and other in italics - it should be unified.
Left side of figure 7 might be outstretched in full width of a page.
Results need proper verification and validation.
Particular well-known parameters are defined in detail whereas some other are just briefly mentioned.
75% of references are of Chinese authors - more references of other authors are suggested. More up-to-date papers are suggested to be referenced.
Figure 12, right part needs discussion since the differences before and after filtering are relatively significant.
Authors might add real-life application of proposed solution. They may suggest it in research such as condition monitoring of railway vehicles and track, e.g. https://doi.org/10.3846/16484142.2017.1342101
Author Response
Response to Reviewers’ Comments for Reviewer #2
Sensors- 492915: Research on Subdivision System of Sin-Cos Encoder Based on Kalman Filtering
Haoning Zhao, Jiazhong Xu*, Haibin Zhang*, Zhen Liu and Shi Dong
Many thanks to the reviewer’s comments and elaborative recommendations and advice on our literature. Such a careful work is made by him/her to improve the quality of this paper and we are quite grateful for the thoughtful and careful work and efforts. All locations of the revisions to the manuscript are highlighted with the YELLOW back color and contents in detail are characterized with red font. The following summarizes our revisions and explanations itemized according to the reviewer’s comments (Note: unless otherwise specified, all the page numbers cited below are for the revised manuscript).
Reviewer’s Comment Point 1:
First paragraph need references in a lot of fragments. So do plenty of other paragraphs.
Authors’ Response 1:
Thank you very much for the comments on the paper. We have made correction according to the Reviewer’s comments. As Reviewer suggested that we have referenced more up-to-date papers as follows:
“1. Dogsa, T.; Solar, M.; Jarc, B. Precision Delay Circuit for Analog Quadrature Signals in Sin/Cos Encoders[J]. IEEE Transactions on Instrumentation and Measurement. 2014, 63, 2795-2803.
2. Chudzikiewicz, A.; Bogacz, R.; Kostrzewski, M.; et al. Condition Monitoring of Railway Track Systems by Using Acceleration Signals on Wheelset Axle-Boxes[J]. Transport. 2018, 33, 30-42.
13. Zhao, L.; Cheng, K.; Chen, S.; et al. An approach to investigate moire patterns of a reflective linear encoder with application to accuracy improvement of a machine tool[J]. Proceedings of the Institution of Mechanical Engineers Part B-Journal of Engineering Manufacture. 2019, 233, 927-936.
15. Pablo, M.; Lai, H.; Oppenheim, A. Delta-Ramp Encoder for Amplitude Sampling and Its Interpretation as Time Encoding[J]. IEEE Transactions on Signal Processing. 2019, 67, 2516-2527.
16. Wu, Q.; Shi, F. A backward approach to certain class of transport equations in any dimension based on the Shannon sampling theorem[J]. International Journal of Computer Mathematics. 2017, 94, 1943-1967.” (Lines 375-407, Page 14 in the revised manuscript)
Reviewer’s Comment Point 2:
Authors should avoid acronyms in titles of chapters and subchapters.
Authors’ Response 2:
We are grateful to the reviewer about the careful work. Considering the Reviewer’s suggestion, we have given the full name in titles of chapters and subchapters.
Reviewer’s Comment Point 3:
Line 69: it should be “It can be observed that…”
Authors’ Response 3:
Sorry about the mistake and thanks to the reviewer’s tip. As Reviewer suggested that “seen” is changed into form of “observed”. (Lines 69, Page 2 in the revised manuscript)
Reviewer’s Comment Point 4:
Shannon sampling theory is given without any references.
Authors’ Response 4:
Thanks for the reviewer’s reminding and the reviewer’s concern is also what we have considered. We have referenced more papers in Shannon sampling theory as follows:
“16. Wu, Q.; Shi, F. A backward approach to certain class of transport equations in any dimension based on the Shannon sampling theorem[J]. International Journal of Computer Mathematics. 2017, 94, 1943-1967.” (Lines 406-407, Page 14 in the revised manuscript)
Reviewer’s Comment Point 5:
Why do Authors use PHI instead of φ?
Authors’ Response 5:
Thank you for the suggestion and sorry for our carelessness here. We have made correction by used “φ” instead of “PHI” according to the Reviewer’s comments. (Lines 157-164, Page 4-5 in the revised manuscript)
Reviewer’s Comment Point 6:
Symbol R is covariance once and later is described as subdivision ratio.
Authors’ Response 6:
Your suggestion is greatly appreciated. We are such sorry about the low-level mistake. We have used E to denote covariance. (Lines 169, Page 5 in the revised manuscript)
Reviewer’s Comment Point 7:
Authors could describe verification (and validation) of the model or method, but not a simulation.
Authors’ Response 7:
Thank you very much for the comments on the paper and the research significance of the study. We have revised the manuscript as suggested and re-organized to highlight the key issues all through the paper as follows:
“In the simulation results, the total rotation angles of the encoder obtained by ATS and APC methods are compared with the theoretical angle. The results show that both methods can achieve a more accurate angle. The maximum error of the APC method is the corresponding angle of the pulse width. However, the physical calculated precision of the encoder can be achieved by the ATS method.
The ATS method is suitable for use in the industrial computer environment, and its accuracy is higher. The APC method has a lower precision than that of the ATS method. However, the APC method is easier to run stably in embedded systems or in systems with stringent real-time requirement in practical engineering applications.”
(Lines 283-291, Page 11 in the revised manuscript)
Reviewer’s Comment Point 8:
Could Authors change “lines” into a symbol?
Authors’ Response 8:
Thanks to the reviewer for the meritorious suggestion. Considering the Reviewer’s suggestion, we have changed “lines” into a symbol “L”.
Reviewer’s Comment Point 9:
Matlab is given in big letters once and later on in small letters.
Authors’ Response 9:
We are grateful to the reviewer about the careful work. The wrong form of “Matlab” is replaced by “MATLAB”.
Reviewer’s Comment Point 10:
Some parameters are given in straight form and other in italics - it should be unified.
Authors’ Response 10:
Many thanks to the reviewer about the valuable suggestion, and we have used italics for all variable.
Reviewer’s Comment Point 11:
Left side of figure 7 might be outstretched in full width of a page.
Authors’ Response 11:
Thank you for the suggestion. As Reviewer suggested that we have narrowed the figure 7. (Lines 251, Page 9 in the revised manuscript)
Reviewer’s Comment Point 12:
Results need proper verification and validation.
Authors’ Response 12:
Thanks to the reviewer for the meritorious suggestion. The reviewer’s question is meritorious and we have also considered it comprehensively. In order to express the point more clearly according to the reviewer’s comments, we add this part as a discussion as follows:
“The peak of the deviation after the filtering is smaller than before the filter, and the overall error decreases. The standard deviation of the filter deviation is smaller than that before the filtering, and the accidental error before the filtering is eliminated. The negative overall tendency of the filtered deviation is due to the phase offset caused by the Kalman filtering algorithm. However, this offset has no effect on the relative rotation angle of the calculation.” (Lines 371-323, Page 12 in the revised manuscript)
“The deviation of the ATS method is one order of magnitude lower than that of the APC method, and the accuracy is relatively higher. The output from the APC method is a digital quantity. This value is affected by the system resolution and generates a pulse system error.
In this paper, a novel subdivision system of the high precision SCE based on a Kalman filter is designed, which can effectively solve the noise problem caused by the hardware and the environment. Two methods are explored: ATS and APC. The ATS method offers high precision; the APC provides a relatively low precision, but offers a fast, stable option for practical, real-time applications.” (Lines 335-343, Page 13 in the revised manuscript)
Reviewer’s Comment Point 13:
Particular well-known parameters are defined in detail whereas some other are just briefly mentioned.
Authors’ Response 13:
The reviewer’s concern is quite significant and we have defined all parameters in detail.
Reviewer’s Comment Point 14:
75% of references are of Chinese authors - more references of other authors are suggested. More up-to-date papers are suggested to be referenced.
Authors’ Response 14:
Thank you very much for the comments. We have revised the manuscript as suggested and re-organized to highlight the key issues all through the paper as follows:
“In the simulation results, the total rotation angles of the encoder obtained by ATS and APC methods are compared with the theoretical angle. The results show that both methods can achieve a more accurate angle. The maximum error of the APC method is the corresponding angle of the pulse width. However, the physical calculated precision of the encoder can be achieved by the ATS method.
The ATS method is suitable for use in the industrial computer environment, and its accuracy is higher. The APC method has a lower precision than that of the ATS method. However, the APC method is easier to run stably in embedded systems or in systems with stringent real-time requirement in practical engineering applications.”
(Lines 283-291, Page 11 in the revised manuscript)
Reviewer’s Comment Point 15:
Figure 12, right part needs discussion since the differences before and after filtering are relatively significant.
Authors’ Response 15:
Thank you very much for the comments on the paper and the research significance of the study. We have re-written this part according to the Reviewer’s suggestion, and we have added to the discussion of the figure 12 right part such as follows:
“Comparing the experimental results of the ATS method and the analogy pulse counting method, the influence of the two software subdivision methods on the accuracy of the calculation of the encoder rotation angle is analyzed. The results of the encoder angle comparison, presented in Figure 13, as follows.
(1) The deviation of the ATS method is one order of magnitude lower than that of the APC method, and the accuracy is relatively higher.
(2) The output from the APC method is a digital quantity. This value is affected by the system resolution and generates a pulse system error.”
(Lines 331-338, Page 13 in the revised manuscript)
Reviewer’s Comment Point 16:
Authors might add real-life application of proposed solution. They may suggest it in research such as condition monitoring of railway vehicles and track, e.g. https://doi.org/10.3846/16484142.2017.1342101
Authors’ Response 16:
Thank you for the suggestion and we have studied the literatures provided by the reviewer systematically. As Reviewer suggested that this paper can use it in research such as condition monitoring of railway vehicles and track. Considering the Reviewer’s suggestion, we have referenced the papers of https://doi.org/10.3846/16484142.2017.1342101 as follows:
“2. Chudzikiewicz, A.; Bogacz, R.; Kostrzewski, M.; et al. Condition Monitoring of Railway Track Systems by Using Acceleration Signals on Wheelset Axle-Boxes[J]. Transport. 2018, 33, 30-42.” (Lines 377-378, Page 14 in the revised manuscript)
At last, The authors would like to thank the reviewer for the constructive review again. We hope that these revisions are some satisfactory. The authors have updated the acknowledgments to express sincere appreciations to the reviewer for the valuable comments to further improve the paper.
Yours sincerely,
Haoning Zhao, Jiazhong Xu, Haibin Zhang, Zhen Liu and Shi Dong

Reviewer 3 Report
The authors provided an additional certificate that confirms proof reading of the manuscript. The language is sound but the content is rather strange. One rather strange example can be found in lines 97-99.
Although the output is a sine analog signal, its original edge needs to enter a high-frequency
excitation signal. - This is rather strange. The encoder provides several output signals that differ in their phases. What is the purpose of the additional excitation signal?
This transformer needs a special decoding circuit and a decoding chip; it only outputs a sin-cos period in a week. - This sentence, although grammatically sound, has no meaning at all. What do you mean by output in a week? Do you mean weak output and not week?
Line 106: What are flow disks, and what is large data file technology?
Line 108: The signal has two-phase difference of 90deg. A single signal can not have phase difference. There should be at least two signals, which are actually generated by the sensor but the presentation is rather cryptic.
Equations (1) and (2) are wrong. The signals in Figure 1 can be described as U_a(t) = sin(\omega t+\phi_a) and U_b(t) = sin(\omega t+\phi_a+\pi/2). They do not have the same phase shift \theta!
Line 123: What is Up?
Line 124: If the input amplitude U > Up , the count pulse is output. - Strange sentence.
Section 3.2 Kalman filter is littered with errors. First, when using KF, you should at least provide the state space representation of your model. I would assume that your model has single state which is the angle \theta, which is rather strange. What is PHI? I assume it is a matrix but of what dimensions and what is its contents.
Line 180: Therefore, the tangent value unification is set to the hypothesis overflow threshold value. - Sorry, but I do not understand this!
Line 182: An SCE instrument error causes the sin-cos signal to exhibit the burr phenomenon. Do you mean this https://en.wikipedia.org/wiki/Burr_(edge) This is rather strange description I suppose the wording is just wrong.
Line 183-185: The text has no meaning whatsoever!
Section 4: The Kalman Filtering from Section 3, does not mention estimation of the frequency of the signal (or at least it is not clearly shown). The question is how the complete method copes with variable rotational frequency. I would assume that the frequency of the generated phase shifted signals is directly related to the rotational speed of the shaft.
Lines 227-229: How did you come up with the values of the noise. As I already assumed, since your noise matrix Q is a scalar, this means that you have just one state in the Kalman Filter.
Section 5. Experimental validation is done for a machine rotating at 300 rpm. This is rather slow moving machine and I can hardly see any issues with tracking such movement even with rather cheap pulse encoders. Please clarify.
Author Response
Response to Reviewers’ Comments for Reviewer #3
Sensors- 492915: Research on Subdivision System of Sin-Cos Encoder Based on Kalman Filtering
Haoning Zhao, Jiazhong Xu*, Haibin Zhang*, Zhen Liu and Shi Dong
The authors are grateful to the reviewer for the positive comments on the contributions of our paper, and the affirmation of our work. The referee’s concern and suggestion are also quite valuable and stimulative to our research and all the comments and suggestions have been carefully considered. The following summarizes our revisions and explanations itemized according to the reviewer’s comments (Note: unless otherwise specified, all the page numbers cited below are for the revised manuscript). All locations of the revisions to the manuscript are highlighted with the YELLOW background color and detailed revision contents are characterized with the red font.
Reviewer’s Comment Point 1:
The authors provided an additional certificate that confirms proof reading of the manuscript. The language is sound but the content is rather strange. One rather strange example can be found in lines 97-99:
Although the output is a sine analog signal, its original edge needs to enter a high-frequency excitation signal. - This is rather strange. The encoder provides several output signals that differ in their phases. What is the purpose of the additional excitation signal?
This transformer needs a special decoding circuit and a decoding chip; it only outputs a sin-cos period in a week. - This sentence, although grammatically sound, has no meaning at all. What do you mean by output in a week? Do you mean weak output and not week?
Authors’ Response 1:
We would like to thank the reviewer for giving us a chance to revise the paper, and also thank the reviewers for giving us constructive suggestions which would help us both in English and in depth to improve the quality of the paper. Sorry about the mistake and thanks to the reviewer’s tip. It is really true as Reviewer suggested that this part of content has no meaning to the theme. So we have removed it in the revised manuscript according to the reviewer’s comment. The following parts in blue font are removed:
“A rotary transformer is an absolute magnetoelectric encoder, which is suitable for harsh environments. Although the output is a sine analog signal, its original edge needs to enter a high-frequency excitation signal. This transformer needs a special decoding circuit and a decoding chip; it only outputs a sin-cos period in a week. The subdivision accuracy of a rotary transformer is limited; it cannot reach the high precision of ordinary incremental encoder. Consequently, an SCE has advantages for high-precision and precision test systems [12].” (Lines 96-101, Page 3 in the revised manuscript)
Reviewer’s Comment Point 2:
Line 106: What are flow disks, and what is large data file technology?
Authors’ Response 2:
Sorry about the confusion that brought by the original manuscript, and we have changed this sentence to read as follows.
“Parallel flow processing technologies of high-speed flow disk and large file data are adopted to achieve this.” (Lines 105-107, Page 3 in the revised manuscript) is changed into form of “Parallel streams processing technologies that the high-speed streams of large file data import disks are adopted to achieve this.” (Lines 105-107, Page 3 in the revised manuscript)
Reviewer’s Comment Point 3:
Line 108: The signal has two-phase difference of 90deg. A single signal cannot have phase difference. There should be at least two signals, which are actually generated by the sensor but the presentation is rather cryptic.
Authors’ Response 3:
We are grateful to the reviewer about the careful work. According to the reviewer’s suggestion, we have rewritten the sentence in the following form to make it express the implication more clearly.
“The sin-cos encoding signal refers to the encoder signal that loads the position information with a sine signal. This signal has a two-phase difference of 90°.” (Lines 108-110, Page 3 in the revised manuscript) is changed into form of “The sin-cos encoding signal refers to the encoder signal that loads the position information with two phase sine signals, which have difference of 90° [17].” (Lines 108-110, Page 3 in the revised manuscript)
Reviewer’s Comment Point 4:
Equations (1) and (2) are wrong. The signals in Figure 1 can be described as U_a(t) = sin(\omega t+\phi_a) and U_b(t) = sin(\omega t+\phi_a+\pi/2). They do not have the same phase shift \theta!
Authors’ Response 4:
Your suggestion is greatly appreciated. We have made correction according to the Reviewer’s comments. The revisions are listed in detail as follows:
“The voltage signals of phase A and phase B can be expressed as follows.
U_a(t) = U sin(\omega t+\phi_a)
U_b(t) = U sin(\omega t+\phi_a+\pi/2)
where U is the SCE output voltage amplitude. \omega is the angular frequency. \phi_a is the initial phase.”
(Lines 123-125, Page 3 in the revised manuscript)
Reviewer’s Comment Point 5:
Line 123: What is Up?
Line 124: If the input amplitude U > Up , the count pulse is output. - Strange sentence.
Authors’ Response 5:
Thanks for the reviewer’s reminding and we are such sorry about the low-level mistake. We have rewritten the sentence in the following form to make it express the implication more clearly.
“The principle of a software subdivision method is to select a voltage reference point U_p in Equation (1) and (2). If the input amplitude U > U_p, the count pulse is output.” (Lines 126-127, Page 4 in the revised manuscript) is replace by “The principle of a software subdivision method can be expressed as follows: by selecting a point voltage U_p in Equation (1) and (2) as the voltage reference point of output counting pulse, the count pulse is output when the input amplitude U > U_p.” (Lines 127-130, Page 4 in the revised manuscript)
Reviewer’s Comment Point 6:
Section 3.2 Kalman filter is littered with errors. First, when using KF, you should at least provide the state space representation of your model. I would assume that your model has single state which is the angle \theta, which is rather strange. What is PHI? I assume it is a matrix but of what dimensions and what is its contents.
Authors’ Response 6:
The reviewer’s suggestion is quite helpful for the implications of the parameters are not such clear. In the original paper, the PHI was the state space representation of our model. Now, we have made correction by used φ instead of PHI to clearly express the state space representation of our model. In order to express the point more clearly according to the reviewer’s comments, we add this part as the expression in section 3.2:
“φ is the parameters of the estimated system, which is 1 in this paper. \delta is system parameter, which is 2*\pi /(4*10^7) in this paper.
φ’ represents the transposed matrix of φ, which is 1 in this paper. Q is the deviation of the system process, which is 4*10^7 in this paper.
H are the parameters of the measured system, which is 1 in this paper. V(k) is the measured noise and its covariance is E. E is 0.1 in this paper.
H’ represents the transposed matrix of H. H’ is 1 in this paper.
I is the identity matrix which is 1 in this paper for single-model and single-measurement system.”
(Lines 160-171, Page 4-5 in the revised manuscript)
Reviewer’s Comment Point 7:
Line 180: Therefore, the tangent value unification is set to the hypothesis overflow threshold value. - Sorry, but I do not understand this!
Authors’ Response 7:
The authors would like to thank the reviewer for the constructive review. To clarify this case, the authors have added the section 5.3 in the text as below:
“Therefore, the overflowing tangent value is set to the overflowing threshold value such as ±1000. Then, the peak values of tangent are ±1000.” (Lines 188-189, Page 6 in the revised manuscript)
Reviewer’s Comment Point 8:
Line 182: An SCE instrument error causes the sin-cos signal to exhibit the burr phenomenon. Do you mean this https://en.wikipedia.org/wiki/Burr_(edge) This is rather strange description I suppose the wording is just wrong.
Authors’ Response 8:
We are such sorry about the low-level mistake. The “burr” is replace by “jitter” in the revised manuscript. (Line 191, Page 6 in the revised manuscript)
Reviewer’s Comment Point 9:
Line 183-185: The text has no meaning whatsoever!
Authors’ Response 9:
Your suggestion is greatly appreciated. So we have removed it in the revised manuscript according to the reviewer’s comment. The following parts in blue font are removed to avoid the redundancy and highlight the key points of this study:
“The zero-crossing jitter, which has a negative impact on the analog pulse count, can be avoided a drop along the trigger threshold. Taking the drop along the trigger threshold for the program to judge is Down=-4900, when T(n-1)*T(n)<Down and T(n-1)<-sqrt(-Down); that is T(n)=-T(n);” (Line 192-195, Page 6 in the revised manuscript)
Reviewer’s Comment Point 10:
Section 4: The Kalman Filtering from Section 3, does not mention estimation of the frequency of the signal (or at least it is not clearly shown). The question is how the complete method copes with variable rotational frequency. I would assume that the frequency of the generated phase shifted signals is directly related to the rotational speed of the shaft.
Authors’ Response 10:
Thanks to the reviewer for the meritorious suggestion. The reviewer’s question is meritorious and we have also considered it comprehensively. According to the current model, the Kalman filter cannot be used in the variable rotational frequency of the signal. To clarify this case, the authors have added this part in the text as below:
“The application scenario of this paper is steady state or approximate steady state, and the speed change can be neglected.”(Line 154-156, Page 4 in the revised manuscript)
Thank you very much for the suggestion. The filtering of variable rotational frequency will be an important problem which we need to solve in the next step. The Kalman filter of variable parameter or the filter of other state can be considered.
Reviewer’s Comment Point 11:
Lines 227-229: How did you come up with the values of the noise. As I already assumed, since your noise matrix Q is a scalar, this means that you have just one state in the Kalman Filter.
Authors’ Response 11:
Many thanks to the reviewer about the valuable suggestion. The reviewer’s concern is also what we have considered. As the reviewer said, the Kalman Filter in this paper has just one state. We design noise matrix Q, because the noise level is controllable in the simulation signal. However, in practical application, the parameters need to be adjusted continuously according to the actual system to obtain the optimal results. We will continue to try new parameter optimization methods to let the system automatically get the value of Q.
Reviewer’s Comment Point 12:
Section 5. Experimental validation is done for a machine rotating at 300 rpm. This is rather slow moving machine and I can hardly see any issues with tracking such movement even with rather cheap pulse encoders. Please clarify.
Authors’ Response 12:
Thank you very much for the comments. As the reviewer said, in the application of control system, it is rare to use high precision encoder to measure the rotation angle at low speed. However, our application is a new scenario: precision measurement of robot reducer. The reducer input and the output of the angle measurement accuracy requirements have reached 1’. At present, according to Japanese standards, the recommended measuring of the input speed is between 100-300 rpm, and the recommended measuring of the output speed is between 1000-3000 rpm. It's not that we use the high precision encoder to obtain real-time speed for feedback control, but it is necessary to be able to accurately obtain the actual rotation angles of the input and output at each time in the middle of the reducer rotation process ( and the accuracy requirements have reached 1”). Thus, it can be calculated whether the measuring accuracy of the reducer is reasonable and whether it is consistent with the standard parameters.
Finally, the authors would like to thank the reviewer again for the constructive comments and hope that the revisions are some satisfactory. We have updated the acknowledgments to express sincere appreciations to the reviewer for the valuable comments to further improve the paper.
Yours sincerely,
Haoning Zhao, Jiazhong Xu, Haibin Zhang, Zhen Liu and Shi Dong

Round 2
Reviewer 2 Report
Authors are used to check again the equation (2) - I think there might be used cosines.
Authors mentioned that φ is the parameter of the estimated system, which is 1 in this paper - please, specify what this parameter stands for.
The sentence construction might be improved: „Therefore, the overflowing tangent value is set to the overflowing threshold value …”
Authors are asked to read the paper carefully before resubmission.
Author Response
Response to Reviewer’s Comments
Sensors- 492915: Research on Subdivision System of Sin-Cos Encoder Based on Kalman Filtering
Haoning Zhao, Jiazhong Xu*, Haibin Zhang*, Zhen Liu and Shi Dong
The authors are grateful to the reviewer for the positive comments on the contributions of our paper, and the affirmation of our work. The referee’s concern and suggestion are also quite valuable and stimulative to our research and all the comments and suggestions have been carefully considered. The following summarizes our revisions and explanations itemized according to the reviewer’s comments (Note: unless otherwise specified, all the page numbers cited below are for the revised manuscript). All locations of the revisions to the manuscript are highlighted with the YELLOW background color and detailed revision contents are characterized with the red font.
Reviewer’s Comment Point 1:
Authors are used to check again the equation (2) - I think there might be used cosines.
Authors’ Response 1:
We would like to thank the reviewer for giving us a chance to revise the paper, and also thank the reviewers for giving us constructive suggestions which would help us both in English and in depth to improve the quality of the paper. Sorry about the mistake and thanks to the reviewer’s tip. It is really true as Reviewer suggested that the equation (2) should be used cosines. So we have rewritten the equation in the following form.
“U_B(t) = U sin(\omega t+\phi_a+\pi/2)” (Lines 116, Page 3 in the revised manuscript) is changed into form of “U_B(t) = U cos(\omega t+\phi_a)” (Lines 116, Page 3 in the revised manuscript)
Reviewer’s Comment Point 2:
Authors mentioned that φ is the parameter of the estimated system, which is 1 in this paper - please, specify what this parameter stands for.
Authors’ Response 2:
Thank you very much for the comments. The reviewer’s question is meritorious and we have also considered it comprehensively. In the Kalman filter, φ is the state space representation of our model.
Reviewer’s Comment Point 3:
The sentence construction might be improved: “Therefore, the overflowing tangent value is set to the overflowing threshold value …”
Authors’ Response 3:
We are grateful to the reviewer about the careful work. We are such sorry about the low-level mistake. According to the reviewer’s suggestion, we have rewritten the sentence in the following form to make it express the implication more clearly.
“Therefore, the overflowing tangent value is set to the overflowing threshold value such as ±1000.” (Lines 178-179, Page 5-6 in the revised manuscript) is replace by “Therefore, the overflowing tangent values are set to the overflowing threshold values. For example, the overflowing threshold values are ±1000 in this paper.” (Lines 178-179, Page 5-6 in the revised manuscript)
Reviewer’s Comment Point 4:
Authors are asked to read the paper carefully before resubmission.
Authors’ Response 4:
Thanks a lot for the reviewer’s careful investigation throughout the whole text. We have learned much from it. The authors have scanned the entire manuscript and tried their best to eliminated grammatical errors and language using mistakes, and further refined the revised manuscript to fulfill the high standards for publication.
Finally, the authors would like to thank the reviewer again for the constructive comments and hope that the revisions are some satisfactory. We have updated the acknowledgments to express sincere appreciations to the reviewer for the valuable comments to further improve the paper.
Yours sincerely,
Haoning Zhao, Jiazhong Xu, Haibin Zhang, Zhen Liu and Shi Dong

Reviewer 3 Report
The authors responded to all of the opened questions. However, the answers of the authors opened even more questions.
Estimation of the phase of a sine signal with known and more importantly constant frequency is a very simple problem. Instantaneous phase is a result of a simple Hilbert transform of the signal.
Ignoring the previous comment, let us assume that one would apply Kalman filter for this task. If the Kalman filter has one state and that is the phase, there is no need for any further processing. The phase is just a result of the Kalman filter. The zero crossings can be estimated and hence the speed.
Another way would be to asses the angular speed with Kalman filter, which should be constant. The estimation will have some variance, however only the mean can be extracted as a point estimate and then apply step 1.
If we ignore all of the above, and taking into consideration that as the authors said: “The application scenario of this paper is steady state or approximate steady state, and the speed change can be neglected.”(Line 154-156, Page 4 in the revised manuscript), one can simply apply zero phase bandpass filter to the signal and then extract the phase using the Hilbert transform without any noise!
So sadly, I am unable to see any options but to repeat my decision from the first round.